# Health Advertising during the Lockdown: A Comparative Analysis of Commercial TV in Spain

**DOI:** 10.3390/ijerph18031054

**Published:** 2021-01-25

**Authors:** David Blanco-Herrero, Jorge Gallardo-Camacho, Carlos Arcila-Calderón

**Affiliations:** 1Facultad de Ciencias Sociales, Campus Unamuno, University of Salamanca, Despacho 416, 37007 Salamanca, Spain; david.blanco.herrero@usal.es; 2Departamento de Comunicación, Facultad de Comunicación y Humanidades, University Camilo José de Cela, 28692 Madrid, Spain; jgallardo@ucjc.edu

**Keywords:** agenda-setting theory, COVID-19 pandemic, health advertising, health communication, lockdown, Spain, advertisements, television

## Abstract

During the lockdown declared in Spain to fight the spread of COVID-19 from 14 March to 3 May 2020, a context in which health information has gained relevance, the agenda-setting theory was used to study the proportion of health advertisements broadcasted during this period on Spanish television. Previous and posterior phases were compared, and the period was compared with the same period in 2019. A total of 191,738 advertisements were downloaded using the Instar Analytics application and analyzed using inferential statistics to observe the presence of health advertisements during the four study periods. It was observed that during the lockdown, there were more health advertisements than after, as well as during the same period in 2019, although health advertisements had the strongest presence during the pre-lockdown phase. The presence of most types of health advertisements also changed during the four phases of the study. We conclude that, although many differences can be explained by the time of the year—due to the presence of allergies or colds, for instance—the lockdown and the pandemic affected health advertising. However, the effects were mostly visible after the lockdown, when advertisers and broadcasters had had time to adapt to the unexpected circumstances.

## 1. Introduction

Almost every aspect of human life has been affected by the SARS-COV-2 pandemic since its declaration in March 2020, and academia has rapidly focused on this unprecedented topic and its effects, causes, and derivatives. This has also happened in the field of communication, with numerous studies published regarding the manner in which information about the virus has been consumed during the pandemic [1] or about the potential spread of “fake news” [2].

In general terms, one of the most relevant branches within the field of communication studies is that which focuses on the effects of television media. Some of the most important and influential theories of the field, such as the cultivation theory [3] or the uses and gratifications theory [4], apply to this area. Although television has lost some of its predominance in recent years due to the rise of online and social media, it is still the most consumed medium in most countries. The Estudio General de Medios [5], the most relevant media audience study in Spain, shows that television is still the most used medium, with a penetration of 85%, superior to the 81.1% penetration of the Internet (these data were collected during the first months of 2020, with fieldwork completed just before the lockdown in Spain in mid-March).

Despite the good Internet connection quality in Spain—the penetration rate [6] and average speed of the Internet [7] in Spain are above the European average—it is not as widespread as television, and its use is not as equally distributed among the population groups, with a large underrepresentation of older age groups. In this context, following the lockdown declared in Spain on 14 March 2020, which confined most of the population to their homes, television consumption rose to record levels in March and April, the most restrictive months of the lockdown [8,9]. Due to this important role played by television, it is relevant to focus on this medium.

Given the significant relevance of TV consumption during the lockdown, and the key role of advertising for the business and communication model of this medium, it is of interest to discover how the lockdown has affected the types of advertisements present in television. In addition, in a context in which health has become a constant presence in the media, it is of interest to study whether the same impact has occurred in advertising, particularly during a period in which the consumption of most non-essential goods and services was not possible. Thus, the main goal of this study was to quantify the presence of health advertising in its different forms within the advertisements broadcasted on commercial television in Spain before, during, and after the lockdown. More specifically, we attempted to analyze the kinds of health products or campaigns that were prevalent during this time.

In this way, we hope to fill the existing knowledge gap regarding advertising during the health crisis because, despite the increase in articles related to COVID-19 and its connection with a broad spectrum of objects of study, no relevant works have focused on advertising in the Spanish setting. Thus, this article focuses on a key element during a period in which the awareness of citizens about health-related issues has become dominant, and in which consumption has plummeted and drastically changed, together with the habits, state of mind, and media consumption of a vast majority of citizens. Additionally, the focus on television advertising can be justified not only by the still superior penetration rate of television in the Spanish media market but also by the decreasing investment of advertisers in television, money that is increasingly going to digital campaigns, which have overtaken television as the main advertising platform [10].

### Contextualization of the Study

Health communication, in its broadest sense, was present in the daily life of citizens during the lockdown and in the days before and the weeks and months afterward. However, this field of study is one of the most relevant and traditional in the area of communication studies [11]. The presence of health content in the media has been broadly studied [12,13]; however, some of the most productive objects of study of this area have been the effects of the media coverage of health issues [14], and those campaigns or strategies to promote particular behaviors or awareness using mass media [15,16]. The use of narrative persuasion in health campaigns is one of the most frequently used approaches among these studies [17].

It is clear that communication during a health crisis is more relevant and intense [18]. As Vaughan and Tinker [19] show, health communication plays an important role during a pandemic, especially if a cooperative public is required. A relevant aspect that has been observed during the SARS-COV-2 pandemic is how an emerging infectious disease is communicated, something that authors such as Holmes [20] have studied for a number of years. Relevant theories or models have been developed to study health communication applied to HIV/AIDS [21] and severe acute respiratory syndrome (SARS) [22,23].

Scholars have also shown significant interest in recent years in the role of media during previous health crises, such as the Ebola epidemic of 2014 [24] or the H1N1 flu in 2009 [25]. In recent years, the interest in social media has grown [26], and many studies related to health and risk communication during the current coronavirus pandemic have focused precisely on these media [27]. Severe acute respiratory syndrome (SARS), a disease belonging to the same family of viruses as COVID-19, showed how electronic media make it possible to rapidly disseminate infectious disease prevention messages [28]. Although television is less present in the current academic scenario, it is nonetheless a highly relevant medium, and its role during the COVID-19 pandemic also has been an object of study [29]. The present article contributes to this line of work, focusing on the medium that, to date, still has the highest penetration rate in Spain.

The other key aspect of the present work is advertising, a relevant field of work within communication. The study of advertising in the media is a highly interesting topic, and several investigations have tackled how advertising affects viewers [30]. In general, the analysis of the content depicted in advertising [31] and its effects [32,33] has a high degree of relevance, especially regarding television.

Regarding the connection between health and advertising, the focus has traditionally been on public service announcements (PSAs) to promote healthy habits [34]. However, several studies have also tried to address the presence of advertisements for medicines or medical products and services [35,36] on television. Following this line of work, the present study attempted to discover the frequency of health-related advertisements during the lockdown in Spain, a period during which health was the central aspect in most citizens’ lives, and in which television consumption grew, but advertising investment decreased.

Given the context outlined above, this study took place under the theoretical framework of the first level of the agenda-setting theory, that is, the capacity of media to set the relevance of topics in the public discourse, which is higher if their media presence is also high [37]. Many previous works have used this theory to study the agenda-setting effects of advertising [38,39], although they have focused mostly on political advertising and the second level of the agenda, and few analyses have been undertaken using the first level of agenda-setting to study other types of advertising. Similarly, agenda-setting has been previously used to analyze the coverage of health crises [40], and previous studies have examined the role of the media during the COVID pandemic from the perspective of the agenda-setting theory [41]. However, these approaches have focused mostly on the media coverage or information rather than on the advertising. This study is focused on the application of agenda-setting to advertising outside of politics and, more specifically, during a health crisis. Therefore, based on the agenda-setting theory and the potential effects that health advertising can have during a health crisis in the public discourse, this study attempted to answer the following research questions:RQ1: What was the proportion of health advertisements broadcasted daily before, during, and after the lockdown by the main Spanish commercial television broadcasters?RQ2: Did the presence of different types of health advertisements on the main Spanish commercial television broadcasters increase during the lockdown compared to the same period in the previous year?RQ3: What types of health advertisements were prevalent before, during, and after the lockdown on the main Spanish commercial television broadcasters?

## 2. Materials and Methods

### 2.1. Sample

Although the types of advertising that television offers are broad, our study focuses on advertisements, that is, brief messages shown in the breaks of television programs and the most common and traditional form of television advertising. Following the definition of the *Ley General de Comunicación Audiovisual* [42], we considered advertisements those messages whose goal is to “promote the supply of goods or services, including real estate, rights and obligations”. In this research, we analyzed the advertisements aired in blocks during, before, or after television programs, excluding sponsorship, telepromotion, advertorials or infomercials, and product placement.

A content analysis was conducted of the advertisements broadcasted by the two main television broadcasters in Spain, representing more than 25% of the market share: Antena 3, owned by Atresmedia group, with 11.2% of accumulated share from January to June 2020, and Telecinco, owned by Mediaset, with 14.4% of accumulated share from January to June 2020, according to data from Kantar Media. La 1, the main public television channel and the third most viewed with 9.6% accumulated share until June 2020 was not considered because it is forbidden by law to broadcast commercial advertisements.

Data were collected in four blocks of 50 days each, previously predefined to answer the research questions of the study. Those blocks were:Lockdown: Since the lockdown was decreed in Spain on 14 March 2020, following the declaration of the state of alarm by the government, until Phase 0 of the de-escalation started on 3 May 2020, and the restrictions started to be progressively eased. This was the most dramatic period, during which the harshest restrictions were applied, and it is central to our analysis because the other periods of the study were designed around this one.Pre-lockdown: Fifty days before the beginning of the lockdown, that is, from 24 January 2020 to 13 March 2020. This includes the weeks before the pandemic was declared by the World Health Organization when the virus was gaining media relevance, but, as expected, without having yet influenced the advertising behavior.Post-lockdown: Fifty days after the de-escalation started, that is, 4 May 2020 to 22 June 2020. This period includes the ending of the state of alarm on 21 June 2020. It includes the weeks in which the restrictions were being eased, and people could start resuming some of their normal activities, reaching the so-called “new normality”.Equivalent dates of 2019: From 14 March 2019 to 3 May 2019, that is, the fifty days included in the central period of analysis. During this period in 2019, however, the SARS-COV-2 had not been detected, thus allowing comparison with a period during which there was no influence of the virus.

The collection of the data took place between 1 June 2020 and 6 August 2020 using the Instar Analytics application, developed by Kantar Media, which is the company that conducts audience measurements in Spain. All of the advertisements broadcasted each day in each of the two studied channels were collected and later classified as explained in the following section, obtaining a total sample of 191,738 advertisements (2107 unique cases).

### 2.2. Instrument and Procedure

In addition to the date and time of transmission, the number of contacts, and the channel on which they were broadcasted, the sample of advertisements was content analyzed following a codebook that adapted the consumption categories defined by the European classification of individual consumption by purpose (ECOICOP), which is used to compile the consumption price index (IPC) in Spain and other European countries. This is an expenses classification including all of the different types of goods and services that can be acquired and donations or other possible money expenditures; thus, every commercial good or service, and the advertisements promoting them, can be allocated within one of the categories. Because advertising mostly aims to promote goods and services that are purchased and paid for by consumers, and due to the wide acceptance of this classification, it was considered an adequate basis for our study. However, two modifications were made to adapt the codebook to our study, as follows:

First, there are public service announcements or institutional advertising that does not have an associated expense—antiracism or antitobacco campaigns, for instance—that are also broadcasted as television advertisements. With the aim of including this type of advertising in the study, it was decided to include them within the last category. Similarly, advertisements that improve the corporate image of a brand, although not directly advertising a good or service, can be allocated within the line of the business of the company; in the case of very diverse companies with several branches of activity, these advertisements were allocated within the “miscellaneous goods and services” category. Thus, the general classification used is as follows, with each advertisement classified exclusively under one of the following 12 categories provided by the ECOICOP:Food and non-alcoholic beverages: includes all food products and non-alcoholic beverages that can be purchased for consumption at home;Alcoholic beverages: includes all alcoholic beverages that can be purchased for consumption at home, including low or non-alcoholic beverages that are generally alcoholic, such as non-alcoholic beer. The original ECOICOP category also includes tobacco, but given the prohibition on tobacco advertising, it was not included in the study;Clothing and footwear: includes clothing materials—including fabrics and accessories such as buttons or sewing threads—garments, accessories, footwear and cleaning, repair and hire of clothing, and footwear;Housing, water, electricity, gas and other fuels: includes real estate acquisition or renting, and repair services or materials (painting, minor plumbing, electricians, etc.), other services relating to the dwelling (gardening, security, etc.), and supplies for the dwelling (heating, electricity, water, gas, fuels, heat energy, etc.). In general, this category includes all products and services for the acquisition or maintenance of a dwelling;Furnishings, household equipment and routine household maintenance: includes all household or garden furniture and furnishings, carpets and other floor coverings, lighting equipment, household textiles (bed linen, curtains, table linen, etc.), household appliances, glassware, tableware and household utensils, tools and equipment for house and garden (electric drills, lawnmowers, alarms, etc.), non-durable household goods (cleaning products, candles, nails, fire extinguishers, etc.). It also includes all rental, repair or associated services and cleaning or maintenance services;Health: This category is further discussed below, but in general terms, it includes health products and medicines, health services and health communication, and PSAs;Transport: includes bicycles and motor vehicles (excluding recreational vehicles such as camper vans or boats) and all services and products for their use, reparation, cleaning, or maintenance (including parking expenses, fuel, driving lessons, GPS). It also includes transportation services, such as buses, trains, taxis, flights, or any other private or public transportation method;Communication: includes postal and parcel delivery services (it does not include Amazon or food delivery apps, for example), and telephone and Internet providers, services, or equipment (including smartphones or bundled telecommunication services, but excluding computers or video-on-demand services);Recreation and culture: includes audiovisual, photographic, and information processing equipment (in general, all technological devices, such as computers or televisions, except for smartphones), and all software, applications and Internet-based services and business that do not specifically apply to any other category (for instance, an Internet-based clothes shop would be included in “clothing and footwear”, but Amazon, with a broader scope, would belong in this category). It also includes major durables for recreation (camper vans, canoes, golf carts, etc.), sports equipment, musical instruments, major durables for indoor recreation (gaming machines, billiard tables, etc.), any other recreational items and equipment (toys, games, celebration articles), and flowers and other garden products. It also includes any repair, maintenance, or complementary service or good applied to the previous goods. It also includes pets and any food, product, or service associated with them. Additionally, it includes all recreational or cultural services and attendance, such as sports events, cinema, theater, concerts, museums, television subscriptions, or any other game of chance, both online and offline. Finally, it includes newspapers, books, and stationery, and package holidays;Education: includes all educational services, including language courses, schools, or universities.Restaurants and hotels: includes catering services, including restaurants, bars, and take-away local food services. Furthermore, includes accommodation services;Miscellaneous goods and services: includes personal care (hairdressing, grooming, hygiene, wellness, etc.), baby or child care (nurseries, babysitters, etc.), personal effects (jewelry, clocks, watches, travel goods, articles for babies, etc.), and counseling, insurance, financial, legal, or funeral services. Similar to the case of tobacco, the original ECOICOP category also includes here services such as prostitution, which, given their illegality, are not advertised and, therefore, were not included in the final codebook.

Second, the health category, the most relevant category for our study, was subdivided into more specific subcategories for a more detailed analysis. It must be noted that the original ECOICOP classification has multiple subcategories within each of its 12 broad categories; this subclassification was also used in the case of the “health” category. Although this already existing subclassification was followed, some adaptations were required to obtain a more detailed comparison of the different types of advertisements:Health insurance: although originally included in “other goods and services”, health insurance was moved into this category so that more detailed observations of this type of service could be conducted. When insurance companies were advertised, focusing on other types of insurance or in a general sense, without a focus on the health aspect, they were left outside the health category;Care services, such as caretakers or residences for ill or old people, were also moved from the “other goods and services” to this category;Institutional communication or PSAs related to health issues, both focused on the coronavirus or on other health aspects, were included in two ad hoc subcategories created for the study: one for the advertising of public institutions and one for the advertising of private institutions;Different types of medicines, not differentiated in the original subclassification, were taken into account in the new codebook to enable the study of how medicines that could be used to counter COVID-19 or other specific illnesses were more or less present during the period of analysis. These subcategories were developed following the classification proposed by the World Health Organization in its List of Essential Medicines [43], together with an exploratory observation and the ECOICOP classification.With the goal of focusing on products particularly associated with COVID-19 prevention, a specific group was also added, in which facemasks, hydrogel or gloves were included.

Based on this process, the advertisements classified in the previous variable as “health” were subclassified into one of the following 27 subgroups. All of the other advertisements were coded as “0” in this classification.

Slimming products, such as drugs and other treatments, excluding food products with a health component, such as anti-cholesterol yogurts;Analgesic and anti-inflammatory products, such as ibuprofen or paracetamol;Antacids, including all types of stomach protectors;Contraceptives, including pills and non-oral forms (such as condoms);Antihistamine products;Antibiotics, anti-fungal, and other anti-infective products;Antipyretic drugs specifically designed to fight fever, excluding products that could have such an effect, but not as the main goal, such as analgesics;Antitussive, mucolytic, or anti-flu products: includes drugs and syrups, and nose-sprays or products such as Vicks VapoRub;Vitamin supplements, including drugs and chemical products, but not food products with a health plus, such as calcium-rich milk;Medical creams and spray, including anti-varicose vein, vaginal, and anti-inflammatory creams and similar;Laxatives and antidiarrheal products;Homeopathic products;Throat lozenges;Sleeping pills and other sleeping treatments;Other drugs and medical or pharmaceutical products, including only non-durable products that could not be allocated in any of the previous categories;Facemasks, gloves, and hydrogels, and similar products specifically recommended to prevent COVID-19;Therapeutic equipment, including glasses, hearing aids, wheelchairs, crutches, stairlifts, etc.;Other medical products, including bandage strips, adhesive dressing, syringes, merbromin, and pregnancy tests;Medical or hospital services, including private doctors, plastic surgery, clinics, etc.;Dental services and clinics;Paramedical services, including blood tests, thermal treatments, rehab, physiotherapy, opticians, oculists, or otolaryngologists when the services and clinics were advertised rather than glasses or hearing aids;Private health insurance;Care homes, including retirement homes for elderly persons, residences for disabled persons, or rehabilitation centers;Residential care and assistance, including home help or daycare for elderly or disabled persons at home;Other medical or aid services, including other health-related services not included in the previous categories, for example, medical credits;PSAs by public institutions;PSAs campaigns by private institutions.

This classification is shown in Table 1. Additionally, all items were recoded into dichotomous variables (dummy variables), with 1 indicating the presence of a category or subcategory, and 0 its absence, thus allowing measurement and comparison of the proportion of health advertisements and of each group within this category during each phase or in each channel.

To verify the reliability of the instrument, a randomly selected subsample of 108 different spots was coded by two independent coders. This subsample is around 5% of the sample of different advertisements (N = 2107); because the majority of the 191,738 advertisements were repetitions that would be equally classified, only the sample of unique cases was used to measure the reliability of the instrument. We used Cohen’s kappa and Krippendorff’s alpha (measured from 0 to 1, being 1 total agreement) and obtained an average of 0.797. This was considered to be adequate because it is above the 0.70 threshold [44,45]. This can be seen more in detail in Table 2.

Once the reliability of the instrument was confirmed, the content analysis was conducted. Subsequently, inferential statistics—mostly one-way ANOVA tests and bivariate correlations—were used to answer the research questions. All of the statistical analyses were conducted in IBM’s SPSS (v. 26, IBM, Armonk, NY, USA). A 0.001 (99.9%) type one error was used in this study for stronger rigor of the inferential tests.

## 3. Results

Before answering the research questions, an exploratory, descriptive analysis was conducted to study the general distribution of the variables. Thus, it was observed that Antena 3 broadcasted more advertisements (101,490, 52.9% of the total) than Telecinco (90,248 advertisements, representing 47.1% of the total). Regarding health advertisements, Telecinco paid significantly more attention to this type of advertisement (M = 0.16, SD = 0.36) than Antena 3 (M = 0.15, SD = 0.36; t (188,302.71) = −3.992, *p* < 0.001, d = 0.02), although Antena 3 still led overall, with 15,201 advertisements about health, compared to the 14,111 of Telecinco, due to the larger number of adverts broadcasted in general on this channel.

Of the four periods of study, the most advertisements were broadcasted during the 50 days before the lockdown (63,707 advertisements, 32.7% of the total) and in the 50 days of 2019 (61,854 advertisements, 32.3%), considerably higher than the lockdown (29,466 advertisements, 15.4% of the total) and the post-lockdown (37,711, 19.7% of the total) periods. The daily distribution does not show strong deviations, but it should be noted that, on 16 March 2020, two days after the lockdown was declared, 2356 advertisements were counted, more than twice the average of 958.69 advertisements per day. It is also relevant to note that after the lockdown period and, in particular, during the lockdown, the total number of broadcasted advertisements was significantly smaller; in the case of the lockdown weeks, the number of advertisements was less than half of those broadcasted in the previous phases.

The general distribution of the sample of advertisements is shown in Table 3 in more detail. It should be highlighted that 43,027 advertisements (22.4%) were about food and non-alcoholic beverages, 29,853 (15.6%) were about leisure and culture, 29,312 ads (15.3%) were health-related, and 49,880 (26.0%) fell in the category of other goods and services. Although food and health are rather specific categories, it is important to keep in mind that leisure and culture includes all mobile apps and websites, and most technological devices (with the exception of smartphones, which fall into communication), whereas other goods and services includes all PSAs not related to health, general advertising for supermarkets or shopping chains, and all products of personal hygiene, jewelry, and perfumery. These specific types of advertising are relatively frequent, which partly explains the significant presence of these two broad categories.

Focusing on health advertising, Table 4 shows that the most common type of advertisement during the 200 days of analysis was the PSAs of public institutions with 8574 advertisements (4.5% of the whole sample), followed by medicinal creams or sprays (3170 advertisements, 1.7% of the sample) and dental services, such as clinics or treatments (3064 advertisements and 1.6% of the sample. In all of the periods of study, no advertisements were shown for antibiotics, anti-fungal and other anti-infective products, antipyretics, homeopathic products, other medical products, or care homes.

### Health Advertising during the Lockdown

To answer RQ1 and RQ2, we recoded the general classification so that we could measure the presence of health advertisements (coded 1) compared to all other categories (coded 0). This allowed us to determine the proportion of health advertising during the lockdown and comparison it with the other analyzed periods. Given that the equality of variances was not assumed, Welch’s F test showed significant differences in the proportion of health advertisements between the four studied periods (F(3, 89,438.41) = 136.26, *p* < 0.001). Dunnett’s T3 test proved that the proportion of advertisements related to health broadcasted during the lockdown (M = 0.16, SD = 0.37) was significantly bigger than the proportion of health advertising in the same period of 2019 (M = 0.14, SD = 0.35; d = 0.06) and in the de-escalation phase (M = 0.13, SD = 0.34; d = 0.08), although it is significantly smaller than in the pre-lockdown phase (M = 0.17, SD = 0.38; d = 0.03). Figure 1 shows the proportion of health advertisements in each phase.

Without taking into account the days of 2019, during which the pandemic played no role, it should be noted that the presence of health advertising decreased significantly with time, with the highest proportion found before the lockdown and the smallest proportion after. This is reinforced by the significant and negative correlation between the day of emission and the presence of health advertisements (R = −0.06; *p* < 0.001), meaning that the proportion of health advertising decreased during these 150 days.

Next, we focused on the health subclassification to compare the proportion of the different types of health advertisements during the different phases, thus answering RQ3. Most types of health advertisements showed significant differences between the four phases. For each subgroup of health products and services, parametric or nonparametric tests were conducted.

The Kruskal–Wallis test was conducted for testing the distribution of slimming products in the different phases, because in one of the periods no advertisements of this kind were found, and non-parametrical tests were required. Significant differences were found (K(3) = 602.82, *p* < 0.001). According to Mann–Whitney *U*, this type of advertisement was found significantly more frequently after (M = 0.01 (The exact mean values of this category are: 2019 (M = 0.0065), pre-lockdown (M = 0.0000), lockdown (M = 0.0005), and post-lockdown (M = 0.0076)), SD = 0.09) than before (M = 0.00, SD = 0.00) and during (M = 0.00, SD = 0.02; d = 0.12) the lockdown. Furthermore, the sample of the previous year (M = 0.01, SD = 0.08) had a significantly greater presence of these advertisements than the pre-lockdown and the lockdown (d = 0.10) phases. These results are shown in Figure 2.

Regarding analgesics and other anti-inflammatory products, the Welch’s test (F(3, 86,101.54) = 30.69, *p* < 0.001) showed that during the post-lockdown period there were significantly more advertisements (M = 0.01 (The exact mean values of this category are: 2019 (M = 0.0046), pre-lockdown (M = 0.0065), lockdown (M = 0.0061) and post-lockdown (M = 0.0101)), SD = 0.10) than during the pre-lockdown (M = 0.01, SD = 0.08; d = 0.04), during the lockdown (M = 0.01, SD = 0.08; d = 0.04), and during the sampled period of 2019 (M = 0.00, SD = 0.07; d = 0.06). During the pre-lockdown there were also significantly more advertisements in this category than in the 2019 period (d = 0.03) (Figure 3).

Welch’s test showed antacids were differently present in the four studied periods (F(3, 87,755.69) = 8.14, *p* < 0.001). As can be seen in Figure 4, and according to Dunnett’s T3 test, before the lockdown there were significantly fewer of these advertisements (M = 0.00 (The exact mean values of this category are: 2019 (M = 0.0048), pre-lockdown (M = 0.0032), lockdown (M = 0.0042), and post-lockdown (M = 0.0044)), SD = 0.06) than after the confinement (M = 0.00, SD = 0.07; d = 0.02) and in 2019 (M = 0.00, SD = 0.07; d = 0.03).

Advertisements of antihistamine products also experienced, according to the Kruskal–Wallis nonparametric test, (K(3) = 79.79, *p* < 0.001), differences in the studied phases. A Mann–Whitney *U* test showed that during the lockdown (M = 0.00 (The exact mean values of this category are: 2019 (M = 0.0011), pre-lockdown (M = 0.0000), lockdown (M = 0.0009), and post-lockdown (M = 0.0003)), SD = 0.03) significantly more adverts of this kind were broadcasted than afterward (M = 0.00, SD = 0.02; d = 0.03) and before (M = 0.00, SD = 0.00). Similarly, the sample of 2019 (M = 0.00, SD = 0.03) also had a significantly greater presence of these advertisements than the pre- (d = 0.03) and post-lockdown phases. These differences can be seen more clearly in Figure 5.

Anti-flu products also showed significant differences according to Welch’s test (F(3, 85,081.00) = 458.04, *p* < 0.001). As can be seen in Figure 6, the proportion during the period of 2019 (M = 0.00 (The exact mean values of this category are: 2019 (M = 0.0068), pre-lockdown (M = 0.0027), lockdown (M = 0.0154), and post-lockdown (M = 0.0044)), SD = 0.04) was significantly smaller than during the lockdown (M = 0.01, SD = 0.09; d = 0.10) and during the pre-lockdown (M = 0.02, SD = 0.15; d = 0.19) phases. In addition, the proportion during the post-lockdown weeks (M = 0.00, SD = 0.03) was significantly smaller than during the lockdown (d = 0.11) and before it (d = 0.20). Finally, the proportion of these advertisements was significantly smaller during the lockdown than before (d = 0.12).

Regarding vitamin supplements, Welch’s test showed also significant differences (F(3, 81,565.62) = 18.03, *p* < 0.001). Dunnett’s T3 test showed differences between all groups, with the lockdown period showing the greatest presence of this type of advertisement (M = 0.02 (The exact mean values of this category are: 2019 (M = 0.0068), pre-lockdown (M = 0.0027), lockdown (M = 0.0154), and post-lockdown (M = 0.0044))., SD = 0.12), and significantly more than the same period in 2019 (M = 0.01, SD = 0.08; d = 0.08), the post-lockdown period (M = 0.00, SD = 0.07; d = 0.11), and the pre-lockdown period (M = 0.00, SD = 0.05; d = 0.13). Similarly, the sample of 2019 had a significantly greater proportion of these advertisements than the post lockdown (d = 0.03) and the pre-lockdown (d = 0.06) phases, whereas the post-lockdown phase had significantly more presence of advertisements about vitamin supplements than the pre-lockdown phase (d = 0.03). This is also shown in Figure 7.

The presence of medical creams and sprays also experienced differences, according to Welch’s test (F(3, 86,697.18) = 26.28, *p* < 0.001), although Dunnett’s T3 test showed that it was only during the pre-lockdown period (M = 0.01 (The exact mean values of this category are: 2019 (M = 0.0174), pre-lockdown (M = 0.0131), lockdown (M = 0.0197), and post-lockdown (M = 0.0182)), SD = 0.11) when there were significantly fewer of these advertisements, compared to the lockdown (M = 0.02, SD = 0.14; d = 0.05), the post-lockdown (M = 0.02, SD = 0.13; d = 0.04), and the 2019 (M = 0.02, SD = 0.13; d = 0.04) periods. Figure 8 shows these differences.

Advertising of laxatives was also significantly different according to Welch’s test (F(3, 81,687.35) = 257.15, *p* < 0.001), and Dunnett’s T3 test showed that during the post-lockdown (M = 0.00, SD = 0.01) significantly fewer laxatives and antidiarrhea advertisements were broadcasted than in the lockdown (M = 0.00 (The exact mean values of this category are: 2019 (M = 0.0083), pre-lockdown (M = 0.0047), lockdown (M = 0.0045), and post-lockdown (M = 0.0002)), SD = 0.07; d = 0.09), the pre-lockdown (M = 0.00, SD = 0.07; d = 0.09), and the sample of 2019 (M = 0.01, SD = 0.09; d = 0.12). Furthermore, the sample of 2019 had a significantly greater presence of these advertisements than the lockdown (d = 0.05) and the pre-lockdown (d = 0.04) periods. This is shown in Figure 9.

It was observed that the presence of throat lozenges also experienced significant changes according to the nonparametric Kruskal–Wallis test (K(3) = 1034.04, *p* < 0.001). As can also be seen in Figure 10, the Mann–Whitney *U* test showed there were significantly more advertisements of this kind before (M = 0.01, SD = 0.10) than during the lockdown (M = 0.00, SD = 0.05; d = 0.10), afterward (M = 0.00 (The exact mean values of this category are: 2019 (M = 0.0004), pre-lockdown (M = 0.0104), lockdown (M = 0.0026), and post-lockdown (M = 0.0000)), SD = 0.00), and the sample of 2019 (M = 0.00, SD = 0.02; d = 0.14). In addition, the lockdown period had a significantly greater presence of these advertisements than the following phase and the same period of 2019 (d = 0.07).

Sleeping products also showed significant differences according to Welch’s test (F(3, 77,618.59) = 140.523, *p* < 0.001), and these differences were significant between all phases according to Dunnett’s T3 test: the greatest presence of these sleeping products could be found during the lockdown (M = 0.01 (The exact mean values of this category are: 2019 (M = 0.0020), pre-lockdown (M = 0.0060), lockdown (M = 0.0114), and post-lockdown (M = 0.0090)), SD = 0.11), which was significantly bigger that afterward (M = 0.01, SD = 0.09; d = 0.02), the pre-lockdown period (M = 0.01, SD = 0.08; d = 0.06) and in 2019 (M = 0.00, SD = 0.04; d = 0.12). Additionally, the post-lockdown phase had significantly more presence of this advertisements than the pre-lockdown (d = 0.03) and the 2019 (d = 0.10) phases, whereas the presence before the lockdown was bigger than in the sample of 2019 (d = 0.06). This is visible in Figure 11.

The miscellaneous remainder of pharmaceutical products also showed significant differences according to Welch’s test (F(3, 88,480.72) = 26.28, *p* < 0.001). Dunnett’s T3 test showed that the pre-lockdown period (M = 0.01 SD = 0.12) was when most advertisements of this kind were found, significantly more than in the lockdown period (M = 0.01 (The exact mean values of this category are: 2019 (M = 0.0072), pre-lockdown (M = 0.0141), lockdown (M = 0.0103), and post-lockdown (M = 0.0089)), SD = 0.10; d = 0.03), the post-lockdown period (M = 0.01, SD = 0.09; d = 0.05), and the 2019 studied phase (M = 0.01, SD = 0.08; d = 0.07). Similarly, during the sample of 2019 there were significantly fewer of these advertisements than during the lockdown (d = 0.03) and the de-escalation phase (d = 0.02). These differences are shown in Figure 12.

Due to the lack of relevance of the phenomenon before the lockdown, no advertisements for masks, gloves, and other products to prevent the spread of COVID were broadcasted, so nonparametric tests were needed. The Kruskal–Wallis test showed significant differences (K(3) = 766.57, *p* < 0.001) among this category, and the Mann–Whitney *U* test showed that after the lockdown there were significantly more advertisements of this kind than in 2019 and before the lockdown (both with no cases; M = 0.00 (The exact mean values of this category are: 2019 (M = 0.0000), pre-lockdown (M = 0.0000), lockdown (M = 0.0002), and post-lockdown (M = 0.0130)), SD = 0.00), in addition to during the lockdown (M = 0.00, SD = 0.01; d = 0.10). This is shown in Figure 13.

Therapeutical products also had a significantly different presence in the different studied phases, as Welch’s test showed (F(3, 99,450.70) = 189.94, *p* < 0.001). Dunnett’s T3 test showed the lockdown (M = 0.00, SD = 0.05) had a significantly lower presence of these advertisements than the pre- (M = 0.01 (The exact mean values of this category are: 2019 (M = 0.0146), pre-lockdown (M = 0.0063), lockdown (M = 0.0020) and post-lockdown (M = 0.0064)), SD = 0.08; d = 0.07), and post-lockdown periods (M = 0.01, SD = 0.08; d = 0.07), and the 2019 period (M = 0.01, SD = 0.12; d = 0.14). Furthermore, the sample of 2019 had also a significantly higher presence than before (d = 0.08) and after (d = 0.08) the lockdown. These results are shown in Figure 14.

Welch’s test also showed significant differences (F(3, 86,697.18) = 26.28, *p* < 0.001) regarding the presence of dental services advertisements. Dunnett’s T3 showed that all phases were significantly different: the lockdown period (M = 0.00 (The exact mean values of this category are: 2019 (M = 0.0238), pre-lockdown (M = 0.0195), lockdown (M = 0.0031), and post-lockdown (M = 0.0075)), SD = 0.06) had a smaller presence of these advertisements than the de-escalation phase (M = 0.01, SD = 0.09; d = 0.06), the pre-lockdown weeks (M = 0.02, SD = 0.14; d = 0.16), and the sample of 2019 (M = 0.02, SD = 0.15; d = 0.18). The sample of 2019 had more presence of these advertisements than before (d = 0.03) and after (d = 0.13) the lockdown. Finally, the proportion of advertisements of dental services was bigger in the pre-lockdown phase than after the lockdown (d = 0.10). This can be observed in Figure 15.

Although less present in general, paramedical services also showed significant differences according to Welch’s test (F(3, 102,707.98) = 47.85, *p* < 0.001). As can be seen in Figure 16, all periods were significantly different from each other: the studied sample of 2019 (M = 0.00 (The exact mean values of this category are: 2019 (M = 0.0025), pre-lockdown (M = 0.0018), lockdown (M = 0.0002), and post-lockdown (M = 0.0009)), SD = 0.05) had a greater presence of these advertisements than the pre-lockdown phase (M = 0.00, SD = 0.04; d = 0.02), the post-lockdown phase (M = 0.00, SD = 0.03; d = 0.04), and the lockdown weeks (M = 0.00, SD = 0.02; d = 0.06). In particular, during the lockdown, the presence of these advertisements was significantly smaller than before (d = 0.05) and afterward (d = 0.03), whereas during the pre-lockdown phase there was a greater proportion of these advertisements than in the post-lockdown weeks (d = 0.02).

Health insurance advertisements showed significant differences according to the Kruskal–Wallis nonparametric test (K(3) = 182.35, *p* < 0.001). A Mann–Whitney *U* test showed that the sample of 2019 (M = 0.00 (The exact mean values of this category are: 2019 (M = 0.0034), pre-lockdown (M = 0.0028), lockdown (M = 0.0005) and post-lockdown (M = 0.0000)), SD = 0.06) had a significantly greater presence than the lockdown (M = 0.00, SD = 0.02; d = 0.07) and the post-lockdown (M = 0.00, SD = 0.00) phases. The same was observed with the pre-lockdown phase (M = 0.00, SD = 0.53), with a significantly greater presence of these advertisements than during (d = 0.06) and after the lockdown. Figure 17 shows this more clearly.

Residential care and assistance advertisements were only present in 2019 and in the post-lockdown phase, so non-parametrical tests were conducted; the Kruskal–Wallis test showed that the differences were significant (K(3) = 173.03, *p* < 0.001); the Mann–Whitney *U* test showed that the post-lockdown period had a significantly greater presence (M = 0.00 (The exact mean values of this category are: 2019 (M = 0.0001), pre-lockdown (M = 0.0000), lockdown (M = 0.0000), and post-lockdown (M = 0.0013)), SD = 0.04) than the sample in 2019 (M = 0.00, SD = 0.01; d = 0.05), in addition to the lockdown and the weeks before, during both of which no advertisements of this kind were broadcast (M = 0.00, SD = 0.00). This is shown in Figure 18.

As shown in Figure 19, similar to the previous case, other medical services were only advertised during the post-lockdown phase (M = 0.00 (The exact mean values of this category are: 2019 (M = 0.0000), pre-lockdown (M = 0.0000), lockdown (M = 0.0000), and post-lockdown (M = 0.0006)), SD = 0.03), which resulted in significant differences according to the Kruskal–Wallis test (K(3) = 87.13, *p* < 0.001), and the Mann–Whitney *U* test confirmed that the only significant differences existed between the post-lockdown phase and the other periods, which did not broadcast any advertisement of this kind (M = 0.00, SD = 0.00).

Public PSAs also showed significant differences according to Welch’s test (F(3, 88,304.88) = 152.35, *p* < 0.001). Dunnett’s T3 test showed that during the lockdown phase (M = 0.05 (The exact mean values of this category are: 2019 (M = 0.0343), pre-lockdown (M = 0.0563), lockdown (M = 0.0528), and post-lockdown (M = 0.0362)), SD = 0.22) there were significantly more advertisements of this kind than in the post-lockdown (M = 0.04, SD = 0.19; d = 0.08) and the 2019 (M = 0.03, SD = 0.18; d = 0.09) phases. This was also observed between the pre-lockdown (M = 0.06, SD = 0.23) weeks and the post-lockdown (d = 0.10) and the 2019 (d = 0.11) periods. This is shown in Figure 20.

The presence of private PSAs also experienced significant changes according to Welch’s test (F(3, 71,001.16) = 265.75, *p* < 0.001). All phases were different from each other according to Dunnett’s T3 test: the greatest presence of these advertisements was observed during the lockdown (M = 0.02 (The exact mean values of this category are: 2019 (M = 0.0005), pre-lockdown (M = 0.0025), lockdown (M = 0.0169), and post-lockdown (M = 0.0086)), SD = 0.13), which was significantly more than during the post-lockdown (M = 0.01, SD = 0.09; d = 0.07), the pre-lockdown (M = 0.00, SD = 0.05; d = 0.15), and the 2019 (M = 0.00, SD = 0.02; d = 0.18) phases. During the sample of 2019, significantly fewer advertisements of these kind were broadcasted than before (d = 0.05) and after (d = 0.12) the lockdown, whereas after the lockdown there was also a significantly greater presence these advertisements than before (d = 0.08). This is shown in Figure 21.

Finally, no significant differences were found regarding the presence of advertisements for contraceptives and medical services. In addition, as already shown, no advertisements for antibiotics and other anti-infective products, antipyretics, homeopathic products, other medical products, or care homes were found in the studied sample.

## 4. Discussion

These results showed the differences in the presence of health advertisements during the 50-day long lockdown in Spain due to COVID-19 in Spring 2020, the previous and subsequent phases, and the same period of 2019. The most relevant observations are the important differences between all phases, which can be explained by the pandemic, but also by the different times of the year, with health issues associated with one or the other. An example of this is the greater presence of health advertisements during the pre-lockdown phase, which could be explained by the fact that this occurred during Spain’s flu season, when colds, for example, are more prevalent. However, the smallest presence of health advertisements during the post-lockdown period seems harder to explain by the seasonal changes; for example, recent studies have observed an increase in joint pain or headaches with higher temperatures [46]. This phenomenon could be better explained by the effects of the pandemic and the lockdown, for instance, by the tiredness of viewers of finding health issues in the media, which resulted in less health advertising, or by the sharp decrease in the presence of health products or services that were not central during the crisis, such as dental or paramedical services and therapeutical products. Similarly, it can be expected that significantly fewer advertisements about health were shown during the 50 days of 2019 compared to the lockdown period when health became a central aspect of peoples’ lives. Nonetheless, the potential effect of the different volume of advertisements in each phase should be noted because a similar number of health advertisements during the lockdown or post-lockdown phases would result in a significantly greater proportion than in the two previous phases, given that the total amount of advertisements is bigger.

Regardless of this general analysis, the fluctuations of this broad category can be explained by how each of the subcategories changed. Thus, the same divergence of motifs can be observed within some of the health subcategories. For example, the presence of antihistamine products, mostly used against spring allergies, was not broadcasted before the lockdown because these weeks took place during winter. For the same reason, it appears reasonable that anti-flu products and throat lozenges were prevalent before the lockdown, during winter months when the “traditional” flu and cold season takes place.

However, some differences appear to be more clearly explained by the effects of the lockdown. Among these, the smaller number of advertisements broadcasted during and after the lockdown can be explained by the decrease in advertising investment due to the complex economic situation. More specifically, focused on the different types of health products and services, we can highlight, for example, how vitamin supplements were prevalent during the lockdown, when citizens were not able to leave their houses, resulting in a lack of sun and deterioration in other healthy habits. Similarly, sleeping products were prevalent during and after the lockdown, perhaps because new schedules and a lack of exercise or work had a negative effect on the sleeping routines of many citizens [47]. The clearest case may be that regarding masks and other COVID-related products, which were mostly present after the lockdown and, to a lesser extent, during the lockdown. Similarly, the strong increase in the PSAs of private companies during the lockdown appears to be clearly related to the fact that companies joined the effort of raising awareness about the importance of staying home, as has been observed in previous studies [48]. In addition, the differences in the presence of advertisements of dental services appear to be consistent with the lockdown because the values during and after the lockdown were far below those of 2019 and before the lockdown, indicating the decrease in interest in this non-COVID19-related health issue. Indeed, dental health was strongly affected by the pandemic because dental procedures were considered to increase the risk of spreading the virus, thus forcing many practitioners to close during the lockdown [49]. These differences also show how the changes in advertising campaigns took place after some time, and the effects were mostly observed after the lockdown, primarily because it took time for companies and broadcasters to adapt to the new situation.

An interesting case is that of slimming products, which were present to a greater extent after the lockdown. This could be due to the gains in weight during this period [50], but also because of the summer season and the desire to have a “beach body”. However, the significant difference between the presence of these adverts between the lockdown, which was close to zero, and the same period of 2019, indicates the potential effect of the lockdown on the reduced desire to have an “attractive” body.

These effects of the pandemic and the lockdown on health advertising showed more attention was paid to some issues, such as COVID prevention, sleeping, or nutrition, than others (for example, dental health). According to the agenda-setting theory, these impacts may have affected the attention paid to the less immediate health issues that were not part of the COVID agenda, with the associated potential risk. Although this effect is not expected to be high, especially in comparison to the agenda effect of news or other media content, more research is needed on how the health advertisements studied in this article affected the visibility of the sickness and health issues they contain among the public.

## 5. Conclusions

In general, it can be concluded that the smallest proportion of health advertisements was found precisely during the lockdown, while the biggest effects produced by the pandemic and the lockdown were visible in the post-lockdown phase, once the de-escalation started. Nonetheless, many of the limitations and the effects of the pandemic continued for many subsequent months. The effect on advertising could be caused by the need for advertisers and broadcasters to adapt to an unexpected situation, for which new advertisements had to be filmed, and new advertising contracts had to be signed. Thus, it would be interesting to continue studying the posterior phases to identify the potential long-term effects. As a hypothesis for future studies, it can be posited that facemasks or other prevention products, strongly connected with the effects of the lockdown and the pandemic, continued to gain presence after 22 June and after the period of study.

Future studies could also investigate another of the relevant conclusions of this work: that the time of the year and the season strongly influence the types of health products and services that are advertised. Thus, for instance, spring-related issues, such as allergies, have a strong influence on the presence of antihistamine products. Similar effects could be expected with slimming products and their increasing presence nearer to the summer season.

To summarize, we conclude that the lockdown had stronger effects on some health-related advertisements, whereas the time of the year had a stronger influence on others. Overall, however, numerous significant differences, both in the presence of health advertisements and in the presence of the different subtypes, indicated a considerable disparity between phases.

Finally, some limitations of the article should be noted, mostly related to the tools used for the study. First, the ECOICOP, as previously mentioned, is a classification developed for consumption rather than advertising. Although the classification was adapted and the health elements that constitute the central aspects of the study were specifically modified, this factor should be taken into account, particularly regarding the exploratory results. It should also be highlighted that some advertisements relate to a whole brand, making it harder to assign them to a specific category; for example, advertisements of insurance companies in which health insurance was not the main subject or supermarkets that advertised the chain rather than a specific product. In these cases, the advertisement was allocated to the “miscellaneous goods and services” (12) category. Finally, it is also relevant to note that drug and medicine advertisements are required to include a disclaimer about the correct use of medicines. This disclaimer is considered in the Instar Analytics application as an independent advertisement and so was classified as a public health PSA, suggesting that caution should be used when analyzing this category. This also helps to explain why this category had the strongest presence among health advertisements. However, no significant effect is expected among other categories because advertisements from within each category follow the same rule, so the proportions in the different phases were not affected by this factor. Similarly, there was no distinction between PSAs relating to COVID-19 and those dealing with other issues. Although a clear predominance of COVID-related PSAs should be expected during the lockdown, future studies are required to further investigate this specific category.

## Figures and Tables

**Figure 1 ijerph-18-01054-f001:**
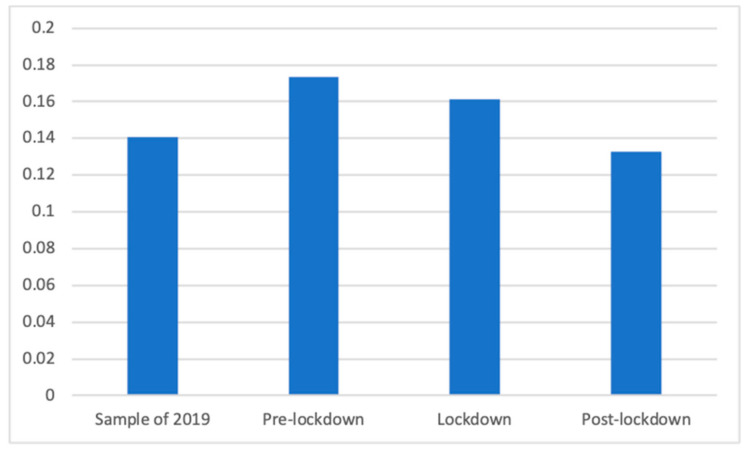
Proportion of health advertisements.

**Figure 2 ijerph-18-01054-f002:**
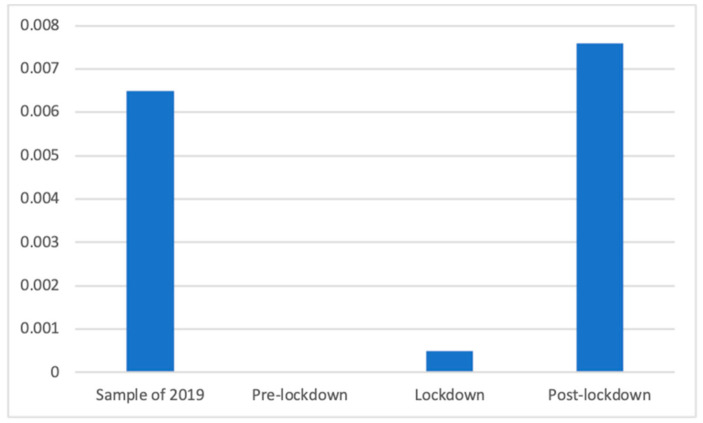
Proportion of advertisements of slimming products in each phase.

**Figure 3 ijerph-18-01054-f003:**
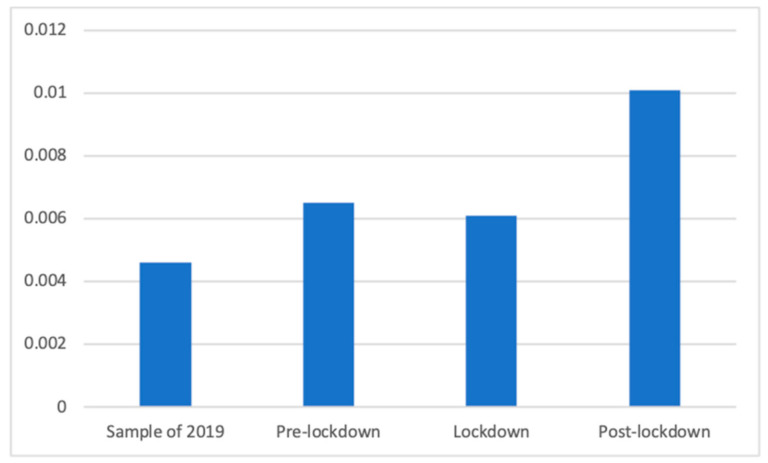
Proportion of advertisements of analgesics and other anti-inflammatory products in each phase.

**Figure 4 ijerph-18-01054-f004:**
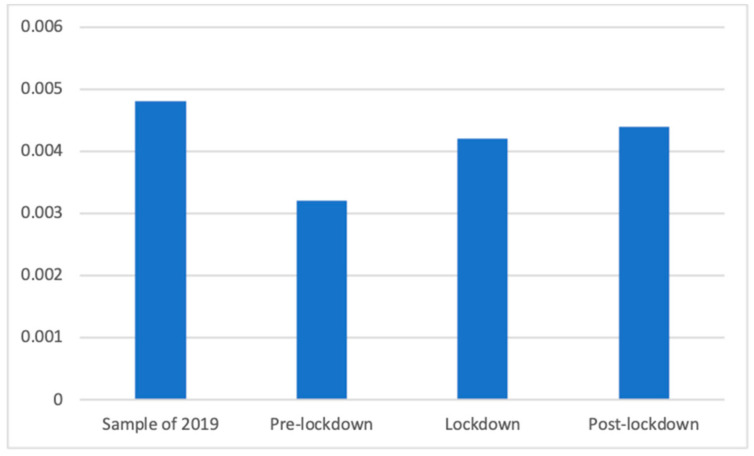
Proportion of advertisements of antacids in each phase.

**Figure 5 ijerph-18-01054-f005:**
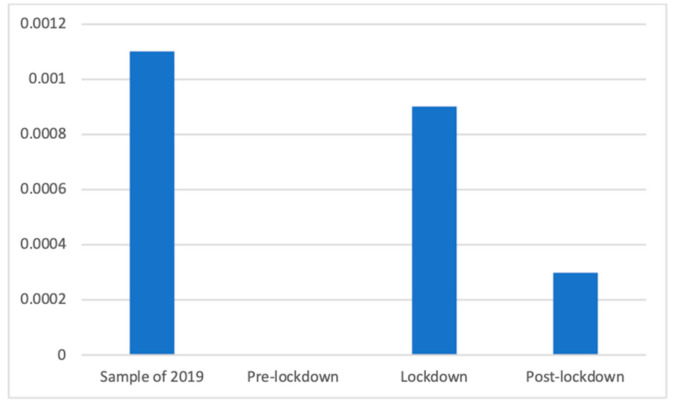
Proportion of advertisements of antihistamine products in each phase.

**Figure 6 ijerph-18-01054-f006:**
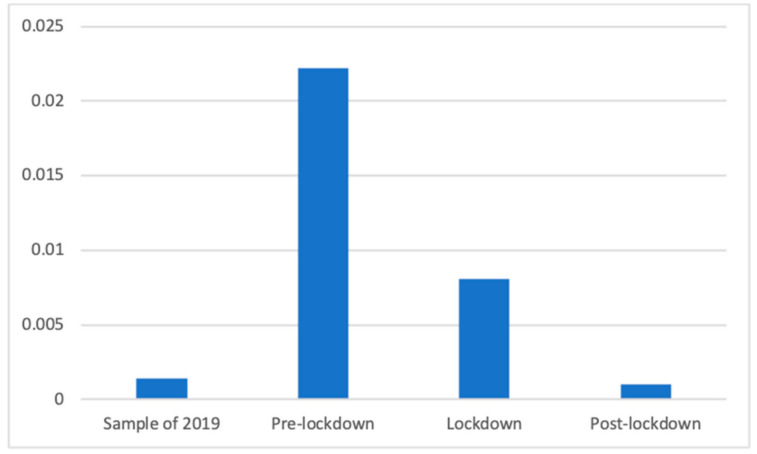
Proportion of advertisements of anti-flu products in each phase.

**Figure 7 ijerph-18-01054-f007:**
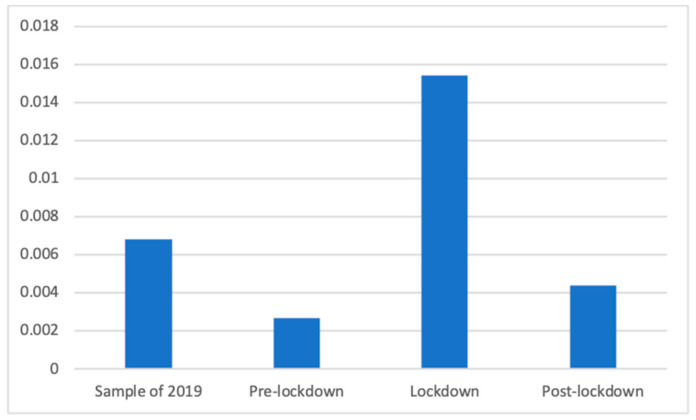
Proportion of advertisements of vitamin supplements in each phase.

**Figure 8 ijerph-18-01054-f008:**
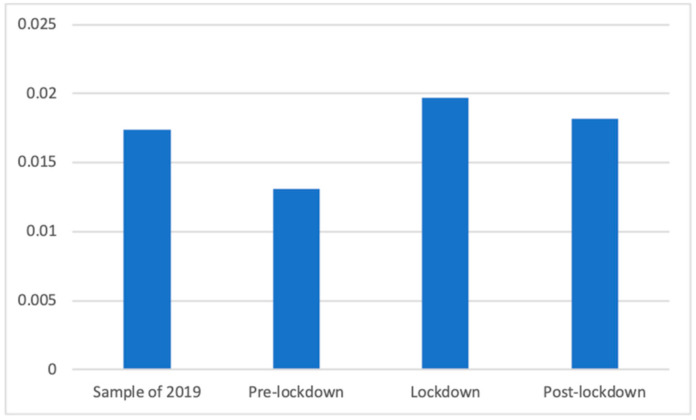
Proportion of advertisements of medical creams and sprays in each phase.

**Figure 9 ijerph-18-01054-f009:**
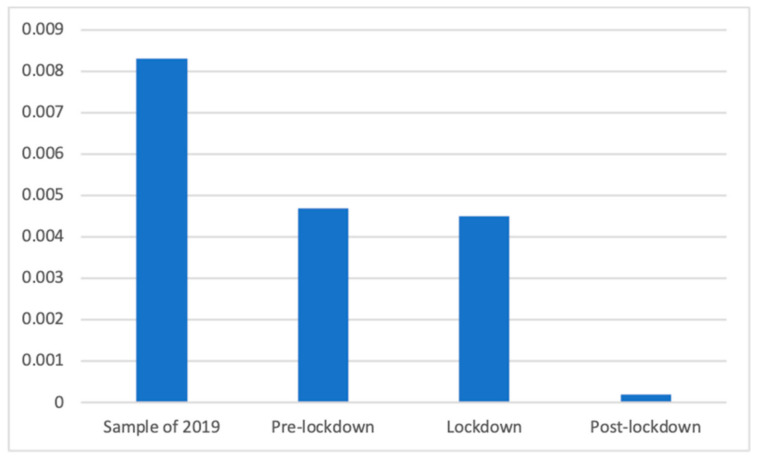
Proportion of advertisements of laxatives and antidiarrheal products in each phase.

**Figure 10 ijerph-18-01054-f010:**
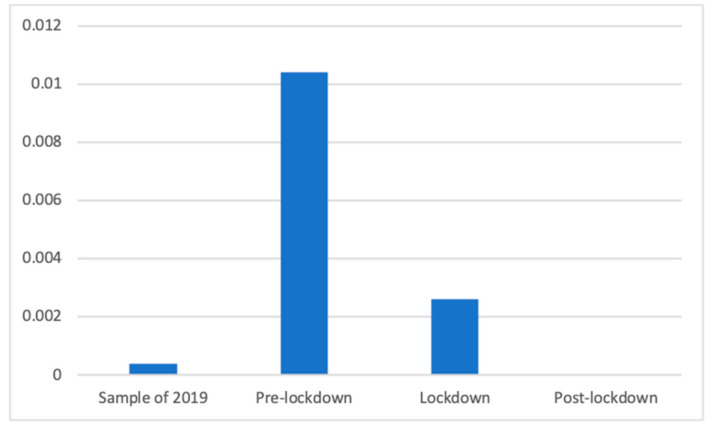
Proportion of advertisements of throat lozenges in each phase.

**Figure 11 ijerph-18-01054-f011:**
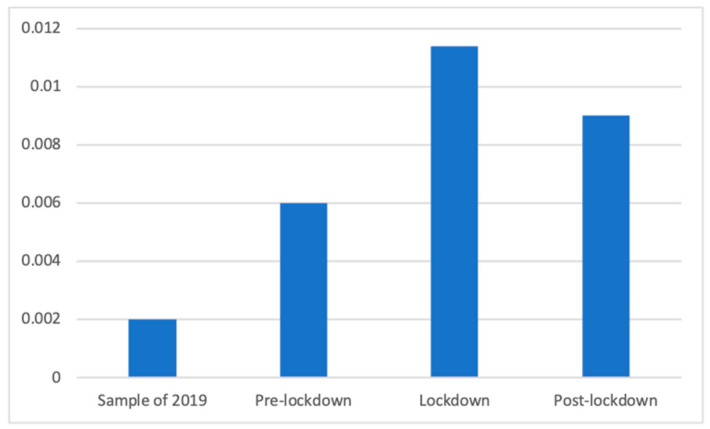
Proportion of advertisements of sleeping products in each phase.

**Figure 12 ijerph-18-01054-f012:**
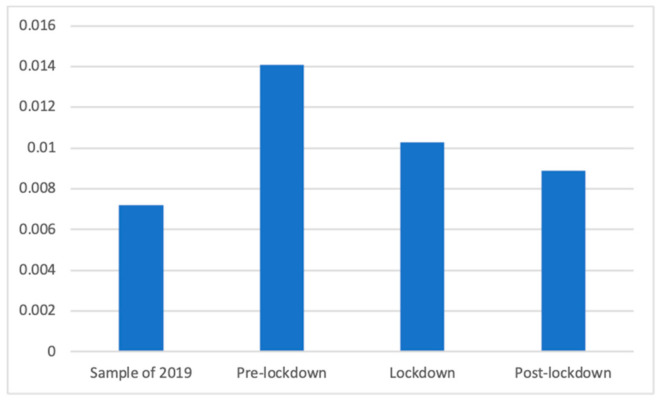
Proportion of the rest of pharmaceutical products in each phase.

**Figure 13 ijerph-18-01054-f013:**
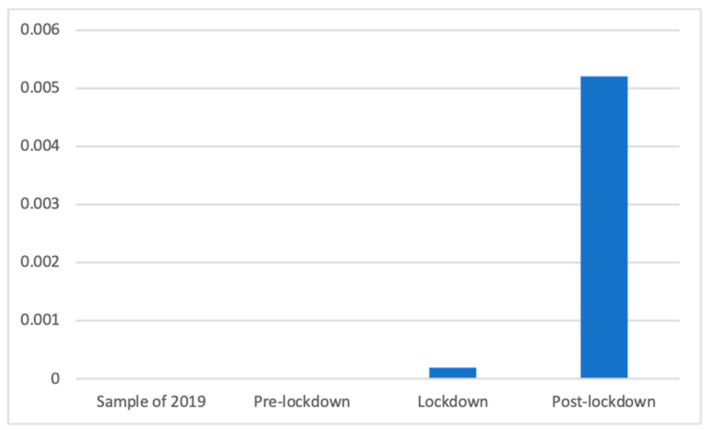
Proportion of advertisements of masks and other anti-COVID products in each phase.

**Figure 14 ijerph-18-01054-f014:**
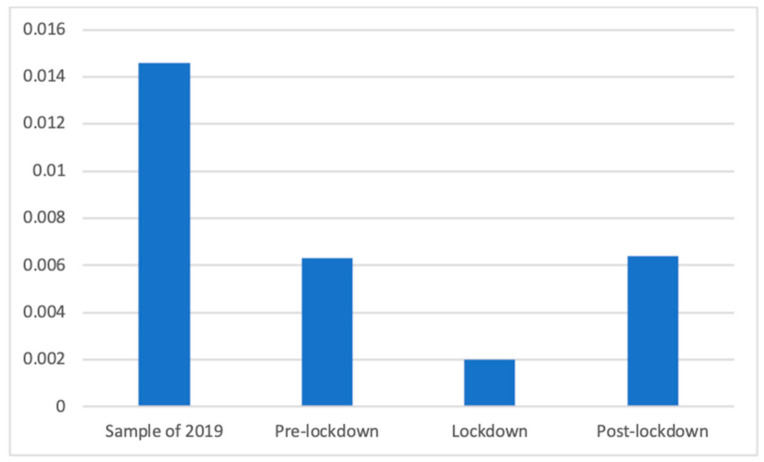
Proportion of advertisements of therapeutical products in each phase.

**Figure 15 ijerph-18-01054-f015:**
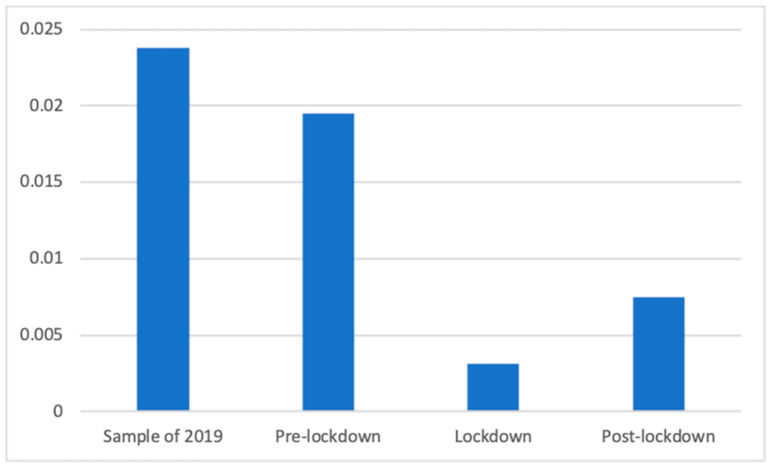
Proportion of advertisements of dental services in each phase.

**Figure 16 ijerph-18-01054-f016:**
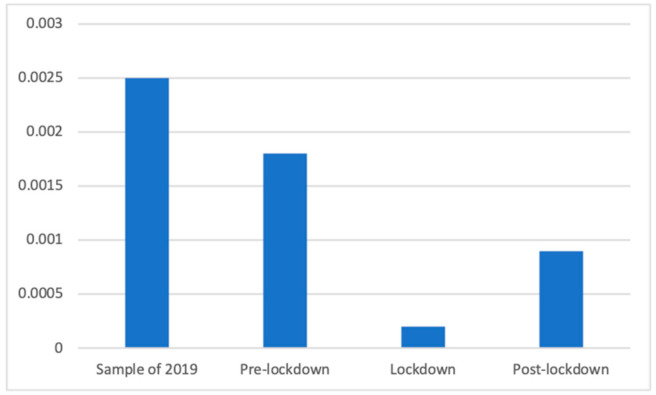
Proportion of advertisements of paramedical services in each phase.

**Figure 17 ijerph-18-01054-f017:**
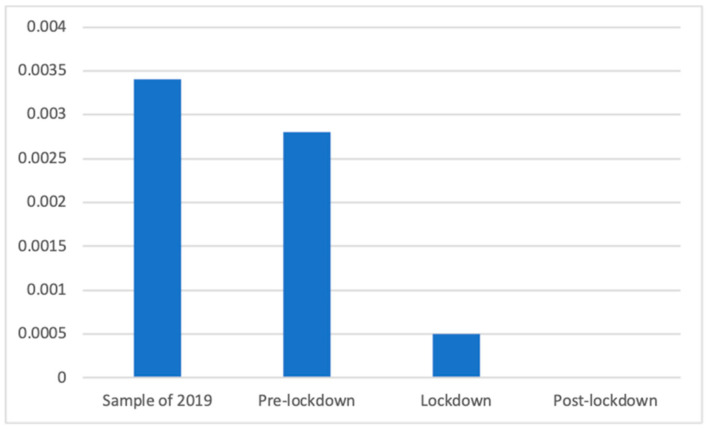
Proportion of advertisements of health insurance in each phase.

**Figure 18 ijerph-18-01054-f018:**
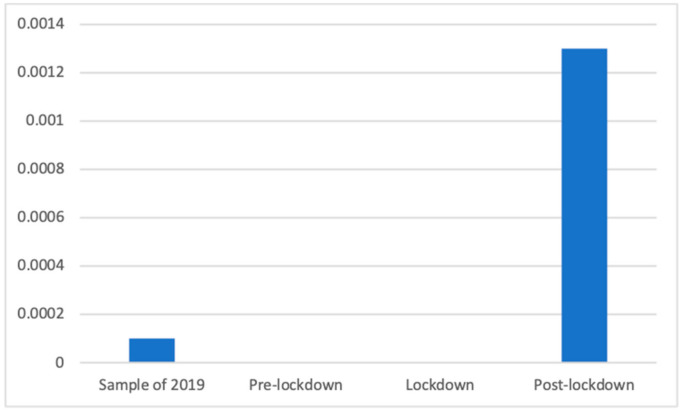
Proportion of advertisements of residential care and assistance in each phase.

**Figure 19 ijerph-18-01054-f019:**
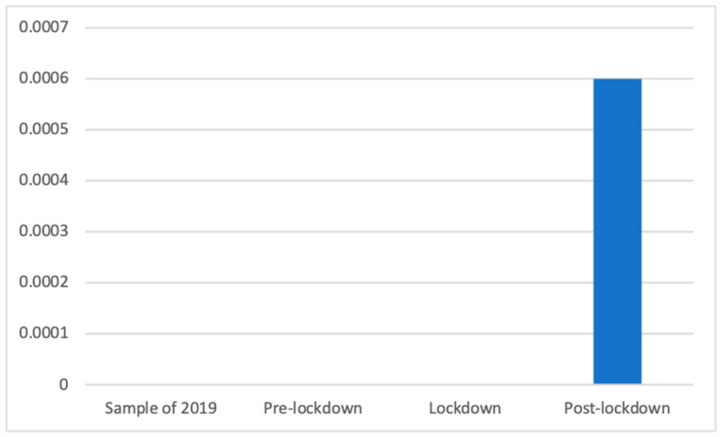
Proportion of advertisements of other medical services in each phase.

**Figure 20 ijerph-18-01054-f020:**
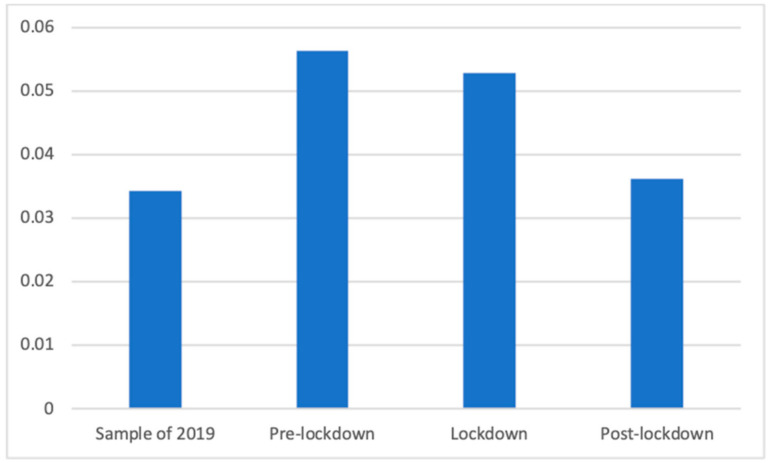
Proportion of advertisements of public PSAs in each phase.

**Figure 21 ijerph-18-01054-f021:**
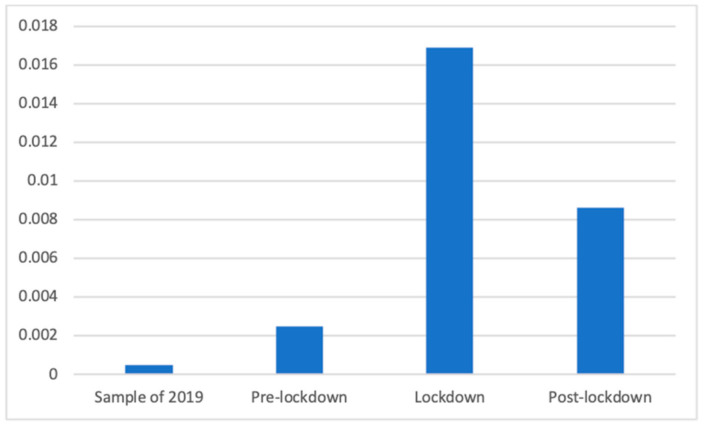
Proportion of advertisements of commercial public service announcements (PSAs) in each phase.

**Table 1 ijerph-18-01054-t001:** Classification of the advertisements.

General Classification	Health Subclassification
Food and non-alcoholic beverages	
Alcoholic beverages	
Clothing and footwear	
Housing, water, electricity, gas and other fuels	
Furnishings, household equipment and routine household maintenance	
Health	Slimming products
	Analgesic and anti-inflammatory products
	Antacids
	Contraceptives
	Antihistamine products
	Antibiotics, anti-fungal and other anti-infective products
	Antipyretic
	Antitussive, mucolytic or anti-flu products
	Vitamin supplements
	Medical creams and sprays
	Laxatives and antidiarrheal products
	Homeopathic products
	Throat lozenges
	Sleeping pills and other sleeping treatments
	Other drugs and medical or pharmaceutical products
	Facemasks, gloves and hydrogels
	Therapeutic equipment
	Other medical products
	Medical or hospital services
	Dental services and clinics
	Paramedical services
	Private health insurance
	Care homes
	Residential care and assistance
	Other medical or aid services
	PSAs by public institutions
	PSAs by private institutions
Transport	
Communication	
Recreation and culture	
Education	
Restaurants and hotels	
Miscellaneous goods and services	

**Table 2 ijerph-18-01054-t002:** Reliability of the measures.

Variable	Cohen’s Kappa	Krippendorff’s Alpha
General classification	0.781	0.781
Health subclassification	0.813	0.813
Average	0.797	0.797

**Table 3 ijerph-18-01054-t003:** General distribution of the sample.

Category	Frequency	Percentage
Miscellaneous goods and services	49,880	26.0
Food and non-alcoholic beverages	43,027	22.4
Recreation and culture	29,853	15.6
Health	29,312	15.3
Furnishings, household equipment and routine household maintenance	13,500	7.0
Transport	8093	4.2
Communication	7511	3.9
Restaurants and hotels	3570	1.9
Housing, water, electricity, gas and other fuels	2378	1.2
Clothing and footwear	2283	1.2
Alcoholic beverages	1672	0.9
Education	659	0.3
Total	191,738	100.0

**Table 4 ijerph-18-01054-t004:** Distribution of health advertisements.

Category	Frequency	Percentage
Non-health advertisements	162,426	84.7
PSAs by public institutions	8574	4.5
Medical creams and sprays	3170	1.7
Dental services and clinics	3064	1.6
Other drugs and medical or pharmaceutical products	1969	1.0
Antitussive, mucolytic or anti-flu products	1756	0.9
Therapeutic equipment	1597	0.8
Analgesic and anti-inflammatory products	1249	0.7
Vitamin supplements	1208	0.6
Sleeping pills and other sleeping treatments	1170	0.6
PSAs by private institutions	1006	0.5
Laxatives and antidiarrheal products	946	0.5
Antacids	786	0.4
Throat lozenges	752	0.4
Slimming products	703	0.4
Private health insurance	403	0.2
Paramedical services	306	0.2
Medical or hospital services	226	0.1
Facemasks, gloves and hydrogels	202	0.1
Antihistamine products	103	0.1
Residential care and assistance	56	0.0
Contraceptives	40	0.0
Other medical or aid services	26	0.0
Antibiotics, anti-fungal and other anti-infective products	0	0.0
Antipyretic	0	0.0
Homeopathic products	0	0.0
Other medical products	0	0.0
Care homes	0	0.0
Total	191,738	100.0

## Data Availability

The data presented in this study are available on request from the corresponding author. The data are not publicly available due to their commercial condition, as they are provided by a private company.

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
