# Peer review of "Health Advertising during the Lockdown: A Comparative Analysis of Commercial TV in Spain"

_ijerph, 2021, doi:10.3390/ijerph18031054_

Round 1

Reviewer 1 Report

This study provides a much-needed look of COVID-19’s influence on the ad placement on TV in a nation hit hard by the pandemic. Advertising is heavily impacted by the outbreak around the world. By focusing on Spain, this study opens up a window to this important international market about how advertising has changed during the lockdown. However, a few caveats should be noted as outlined below.

  • Although the authors claimed that the study was framed by agenda-setting theory, there is few theoretical connections from and to the theory in lit review, research question development, or finding discussion. There is virtually no mention of the theory after page 3. I would like to see the argument of how the theory guides the research idea development, how it leads to the RQs, and how the findings either confirm or dispute with the theory.
  • I admire the authors categorization system but disagree with the use of unnecessary descriptions and result reporting format.
    • Many of the category descriptions are unnecessary and unrelated to the central research topic. As the study is focused on COVID-19 related advertisements, I see no need to mention or even analyze the other categories other than COVID-19 related health ads. Such descriptions can be moved to appendix to save space.
    • Even under the Health subclassification, there are many categories unrelated to COVID-19 and should be either removed or moved to appendix. Such include slimming products, analgesic and anti-inflammatory, antiacids, contraceptives, antihistamine, etc. In fact, the only few categories I see appropriate to the research is facemasks, medical or hospital services, paramedical services, private health insurance, residential care and assistance, and the “awareness campaigns”.
    • A term we used in the US context about “awareness campaigns” is public service announcement or PSA. It can be advertisements sponsored by government agencies, non-profits, or commercial entities. I suggest the authors adopt this term instead of “awareness campaigns”.
    • Regarding the “awareness campaigns”, are those COVID-19 related? It’d be interesting to see the pattern of government/private entity sponsored PSAs about COVID-19 during the lockdown time period.
    • Regarding the reporting format, here are a few suggestions. First, data presented in Table 1/Figure 1 might be relevant so long as the result sheds light on the key research questions. For example, how did health ads’ frequency compare to other types of ads before, during and after the lockdown. Adding the comparison across time blocks help add depth to this finding. I’d suggest removing Table 1 as it is essentially same as Figure 1 and text is harder to read compared to graph.
    • Second, the results of the irrelevant health subclassifications are unnecessary and confusing to read while adding significant length to the paper. I suggest the authors to focus on the few COVID-19 related categories outlined in my comment above.
    • Third, the figures can be combined into one using different color codes to represent the COVID-19 related categories. For example, facemasks can be blue, medical services can be red, and so on.
  • With the slim down result section, the discussion section can be more focused on the relevant COVID-19 impacts, incorporating agenda-setting theory.
  • A grammatical proofreading is needed. Some obvious typos: line 193-194, “the line of business the company”; line 352, “given than”; line 369, “that Telecinco”.
  • Please be consistent with the data reporting about decimal places. A general approach is to use two digits after the decimal, e.g., M = .15 instead of M = 0.1498. Please make this consistent across all statistics, including standard deviations, t statistic, p value, etc.

Author Response

Review 1

We are pleased to re-submit a revised version of the manuscript “Health advertising during the lockdown: A comparative analysis of commercial TV in Spain”. We carefully addressed all the reviewers’ suggestions.

In the following paragraphs, we summarized the modifications made according to each reviewer suggestion.

Reviewer 1:

Although the authors claimed that the study was framed by agenda-setting theory, there is few theoretical connections from and to the theory in lit review, research question development, or finding discussion. There is virtually no mention of the theory after page 3. I would like to see the argument of how the theory guides the research idea development, how it leads to the RQs, and how the findings either confirm or dispute with the theory.

The argumentation of how this theory justifies the RQs have been added in lines 126-127, while a discussion about the connection of this theory with the finding of the study was added in lines 693-700.

I admire the authors categorization system but disagree with the use of unnecessary descriptions and result reporting format. Many of the category descriptions are unnecessary and unrelated to the central research topic. As the study is focused on COVID-19 related advertisements, I see no need to mention or even analyze the other categories other than COVID-19 related health ads. Such descriptions can be moved to appendix to save space. Even under the Health subclassification, there are many categories unrelated to COVID-19 and should be either removed or moved to appendix. Such include slimming products, analgesic and anti-inflammatory, antiacids, contraceptives, antihistamine, etc. In fact, the only few categories I see appropriate to the research is facemasks, medical or hospital services, paramedical services, private health insurance, residential care and assistance, and the “awareness campaigns”.

We consider that all the general categories are necessary for contextualization; similarly, the analysis of all the health subcategories is central to the study, given that we do not study the effects of Covid, but the ones of the lockdown, and this has affected categories such as slimming or sleeping products that have no direct relation with the Covid disease. Therefore, although we agree that a more reduced presentation would be welcome for simplicity, we consider essential to keep all categories in the study, as part of the discussion also focuses on the potential effects of the time of the year, and this comparison is only possible if all the categories are present.

A term we used in the US context about “awareness campaigns” is public service announcement or PSA. It can be advertisements sponsored by government agencies, non-profits, or commercial entities. I suggest the authors adopt this term instead of “awareness campaigns”.

We appreciate the suggestion and have adopted the recommended term throughout the text.

Regarding the “awareness campaigns”, are those COVID-19 related? It’d be interesting to see the pattern of government/private entity sponsored PSAs about COVID-19 during the lockdown time period.

This is not possible, given that the platforms do not offer access to the spots themselves, but only to the brand and type of spot. However, we agree with the interest that this would have on the analysis, so it has been referred in the discussion of results (lines 674-677) and it has been added as a limitation of the study that could be tackled in future studies with a different design (lines 741-744).

Regarding the reporting format, here are a few suggestions. First, data presented in Table 1/Figure 1 might be relevant so long as the result sheds light on the key research questions. For example, how did health ads’ frequency compare to other types of ads before, during and after the lockdown. Adding the comparison across time blocks help add depth to this finding. I’d suggest removing Table 1 as it is essentially same as Figure 1 and text is harder to read compared to graph.

We assume the reference to Table 1 actually refers to Table 3, as it is the one matching what is presented in Figure 1. We consider that the data presented in that table have just an exploratory and context value for the study, and the comparison of the frequency of health spots with other types of ads proposed by this reviewer would open the scope of the article too much, making it loose focus on the object of study, which is the evolution of the proportion of health spots. Regarding the last aspect of the suggestion about removing Table 3 and keeping Figure 1, and given its incompatibility with the recommendation of reviewer 1 of removing the figure and keeping the table, we decided to follow the suggestion of reviewer 1, as we consider that a table allows a more detailed approach to the data and the figure had only a complementary role.

Second, the results of the irrelevant health subclassifications are unnecessary and confusing to read while adding significant length to the paper. I suggest the authors to focus on the few COVID-19 related categories outlined in my comment above.

We have addressed this issue in a previous comment.

Third, the figures can be combined into one using different color codes to represent the COVID-19 related categories. For example, facemasks can be blue, medical services can be red, and so on.

As previously explained, all the categories need to be presented, which makes it impossible to introduce all of them in a comprehensible and manageable figure.

With the slim down result section, the discussion section can be more focused on the relevant COVID-19 impacts, incorporating agenda-setting theory.

As previously indicated, we have added a stronger discussion about agenda-setting theory in lines 693-700.

A grammatical proofreading is needed. Some obvious typos: line 193-194, “the line of business the company”; line 352, “given than”; line 369, “that Telecinco”.

We have fixed these three typos.

Please be consistent with the data reporting about decimal places. A general approach is to use two digits after the decimal, e.g., M = .15 instead of M = 0.1498. Please make this consistent across all statistics, including standard deviations, t statistic, p value, etc.

We have made this change and all the statistical data have been reported using a consistent approach with two decimals (except in the case of the p value, which had been fixed in .001). This made it necessary to report some more specific values in footnotes, given that more than 2 decimals were needed for some very small means. As already explained, after following the recommended system, the article was submitted for professional proofreading and the “0” was returned to all the results reported, so we decided to keep it.

Reviewer 2 Report

This paper analyzes the presence of health spots on television in Spain before, during and after the lockdown caused by the pandemic.
It is a phenomenon of potential interest to readers of the journal, especially those interested in the field of health communication.
In the theoretical framework, it reviews several previous studies related to the content of the paper and justifies the interest of the research according to the agenda-setting theory.
The methods are correct and analysis seem rigorously conducted. Overall it is a good paper.
Here are some recommendations for making changes, most of them regarding the data presentation and the discussion section.

1. Order and structure of the text.
In the methods section a change in the order of presentation of the information is proposed.
It is proposed to relocate lines 196-223 after the current line 293.
Thus, first the general categories of spots are presented, then each one of them is explained.
And secondly, the modifications made in the health subcategory are presented and these subcategories are explained below.

2. Health subclassification.
Is there a reason why the numbers 19, 20 and 28-30 were not used in the health subclassification?
This issue should be explained.

3. Lines 375-378.
"From the four periods of study, it was during the 50 days before the lockdown (63,707 375 spots, a 32.7% of the total) and in the 50 days of 2019 (61,854 spots, 32,3%) when most spots were broadcasted, well over the lockdown (29,466 spots, making 15.4% of the total) and the post-lockdown (37,711, a 19.7% of the total) periods".
Although this is not the aim of the study, it should be noted that during the lockdown and the post-lockdown the number of spots decreased significantly (about half that in previous periods).
A statistical analysis of these differences should be offered.
Also, in the discussion, this data should be mentioned, giving an explanation that, without being too detailed, should be mentioned. For example, it is possible that it was due to the economic impact of the pandemic that reduced the income of some advertisers and therefore their ability to pay for TV spots.

4. Tables.
Tables 3 and 4.
It is recommended to order the categories of spots reflected in each table according to its number of spots (firs the most frequent category and last the least frequent category). In this way, it will be much easier for the reader to draw conclusions about its content.

5. Table 3 / figure 1.
Table 3 and Figure 1 have identic content. It is recommended to delete figure 1, since the table includes this same information with exact data.

6. Table 4 / figure 2.
This table and figure are in the same situation as the previous case.

7. Figures 3-23.
It is recommended to order the columns chronologically. That is, first the period of 2019, and then the pre-lockdown, the lockdown and the post-lockdown. This seems a more logical order that makes it easier to interpret the figure.

8. Data presentation.
In the current version of the paper, 23 figures are presented. This number of figures seems disproportionate.
The authors are suggested to present the data on Figures 4-23 in a single table that allows the results of all the subcategories of health spots to be viewed together.
This table could include the M and SD values of each period, the values of the parametric or non-parametric statistic, the degrees of freedom, the p-value, the effect size and the pairwise comparisons to determine between which periods the differences occurred.
Regarding this last column, for example, in the case of "slimming products" it would be indicated: "post> 2019> lock> pre".
In the case of "analgesics and other anti-inflammatory products", it would be indicated: "post> pre, lock, 2019; pre> 2019".
Only in the cases of the products that the authors consider most relevant from the theoretical point of view, these data could be represented in a figure (although it would be a repetition of the information in the table).

9. Statistical analysis.
The effect size values of the Welch's tests and the Mann-Whitney's U tests should be indicated.

10. Lines 618-627.
The explanations provided for these results must be supported by bibliographic references. For example, to argue that during the winter months there is more publicity about health products, bibliographic references should be sought on the seasonality of this type of spots.

11. Discussion.
In general, the discussion does not truly "discuss" the results with the conclusions of similar previous studies. Interpretations are only offered, with more or less theoretical support, on some of the results.
The discussion should be reformulated so that the results of this study are related to those of the field of study.

12. Formal issues.
1. Keywords must be in alphabetic order.
2. When "p" and "d" values are reported, it mustn't be indicated the entire part of the number. So, it must be reported p< .001; not p< 0.001.
3. Line 352. Erratum "spo(r)ts".

Author Response

We are pleased to re-submit a revised version of the manuscript “Health advertising during the lockdown: A comparative analysis of commercial TV in Spain”. We carefully addressed all the reviewers’ suggestions.

In the following paragraphs, we summarized the modifications made according to each reviewer suggestion.

Reviewer 2:

  1. Order and structure of the text. In the methods section a change in the order of presentation of the information is proposed. It is proposed to relocate lines 196-223 after the current line 293.
    Thus, first the general categories of spots are presented, then each one of them is explained.
    And secondly, the modifications made in the health subcategory are presented and these subcategories are explained below.

This has been modified, and the former lines 196-223 are now lines 265-290.

  1. Health subclassification. Is there a reason why the numbers 19, 20 and 28-30 were not used in the health subclassification? This issue should be explained.

The code of the different variables tries to differentiate between products (OX and 1X), services (2X) and awareness campaigns (3X). However, as this was not relevant for the results and was just used during the classification as an orientation, we removed all the numbers in the Methodology section for more clarity: this affects the list in lines 203-264, the list in lines 294-335 and Table 1.

  1. Lines 375-378. "From the four periods of study, it was during the 50 days before the lockdown (63,707 375 spots, a 32.7% of the total) and in the 50 days of 2019 (61,854 spots, 32,3%) when most spots were broadcasted, well over the lockdown (29,466 spots, making 15.4% of the total) and the post-lockdown (37,711, a 19.7% of the total) periods". Although this is not the aim of the study, it should be noted that during the lockdown and the post-lockdown the number of spots decreased significantly (about half that in previous periods). A statistical analysis of these differences should be offered. Also, in the discussion, this data should be mentioned, giving an explanation that, without being too detailed, should be mentioned. For example, it is possible that it was due to the economic impact of the pandemic that reduced the income of some advertisers and therefore their ability to pay for TV spots.

The information has been broadened in lines 375 to 379 and in the Discussion in lines 665-668.

  1. Tables. Tables 3 and 4. It is recommended to order the categories of spots reflected in each table according to its number of spots (firs the most frequent category and last the least frequent category). In this way, it will be much easier for the reader to draw conclusions about its content.

We have made this change of order in Tables 3 and 4.

  1. Table 3 / figure 1. Table 3 and Figure 1 have identic content. It is recommended to delete figure 1, since the table includes this same information with exact data.

We have removed Figure 1

  1. Table 4 / figure 2. This table and figure are in the same situation as the previous case.

We have removed Figure 2.

  1. Figures 3-23. It is recommended to order the columns chronologically. That is, first the period of 2019, and then the pre-lockdown, the lockdown and the post-lockdown. This seems a more logical order that makes it easier to interpret the figure.

We have changed the order of the columns in these figures, which are now figures 1-21.

  1. Data presentation. In the current version of the paper, 23 figures are presented. This number of figures seems disproportionate. The authors are suggested to present the data on Figures 4-23 in a single table that allows the results of all the subcategories of health spots to be viewed together. This table could include the M and SD values of each period, the values of the parametric or non-parametric statistic, the degrees of freedom, the p-value, the effect size and the pairwise comparisons to determine between which periods the differences occurred.
    Regarding this last column, for example, in the case of "slimming products" it would be indicated: "post> 2019> lock> pre". In the case of "analgesics and other anti-inflammatory products", it would be indicated: "post> pre, lock, 2019; pre> 2019". Only in the cases of the products that the authors consider most relevant from the theoretical point of view, these data could be represented in a figure (although it would be a repetition of the information in the table).

We appreciate the comment, but we believe that the figures are needed so that the text can be clearer and so that each health category can be more easily found and organised. Removing the figures would mean a very long table and a long text without break in which data are hard to find, decreasing also readability. However, we have addressed the previous suggestions of this reviewer, and we believe that the change in the order of the columns introduces more clarity, while the first two figures have been removed, reducing the total amount of figures.

  1. Statistical analysis. The effect size values of the Welch's tests and the Mann-Whitney's U tests should be indicated.

We have added the effect sizes for all significant differences in the Results section, always using Cohen’s d. Only in the paired comparisons in which one of the variables had no values (M = 0.00, SD = 0.00), no effect size was reported given the impossibility of its calculus.

  1. Lines 618-627. The explanations provided for these results must be supported by bibliographic references. For example, to argue that during the winter months there is more publicity about health products, bibliographic references should be sought on the seasonality of this type of spots.

The paragraph has been reformulated so that a reference has been added to the discussion in lines 641-647, trying also to clarify that the ideas expressed in it were just potential hypotheses rather than existing theories.

  1. Discussion. In general, the discussion does not truly "discuss" the results with the conclusions of similar previous studies. Interpretations are only offered, with more or less theoretical support, on some of the results. The discussion should be reformulated so that the results of this study are related to those of the field of study.

The discussion has been improved and broadened, including some new references. Additions to the discussion can be found in lines: 641-647, 651-658, 665-668, 674-677, 681-683 and 693-700. This includes references 46, 48 and 49.

  1. Formal issues. 1. Keywords must be in alphabetic order. 2. When "p" and "d" values are reported, it mustn't be indicated the entire part of the number. So, it must be reported p< .001; not p< 0.001. 3. Line 352. Erratum "spo(r)ts".

These first and the third changes have been made. However, after following the recommended system, the article was submitted for professional proofreading and the “0” was returned to all the results reported, so we decided to keep it.